# Association between Adherence to the Healthy Food Pyramid and Breast Milk Fatty Acids in the First Month of Lactation

**DOI:** 10.3390/nu14245280

**Published:** 2022-12-11

**Authors:** David Ramiro-Cortijo, Gloria Herranz Carrillo, Andrea Gila-Diaz, Santiago Ruvira, Pratibha Singh, Cheyenne Braojos, Camilia R. Martin, Silvia M. Arribas

**Affiliations:** 1Department of Physiology, Faculty of Medicine, Universidad Autónoma de Madrid, C/ Arzobispo Morcillo 2, 28029 Madrid, Spain; 2Division of Gastroenterology, Beth Israel Deaconess Medical Center, Harvard Medical School, 330 Brookline Avenue, Boston, MA 02215, USA; 3Division of Neonatology, Hospital Clínico San Carlos, Instituto de Investigación Sanitaria del Hospital Clínico San Carlos (IdISSC), C/ del Profesor Martin Lagos, S/N, 28040 Madrid, Spain; 4Department of Agricultural Chemistry and Food Science, Faculty of Science, Institute of Food Science Research (CIAL, UAM-CSIC), Universidad Autónoma de Madrid, C/ Nicolas Cabrera 9, 28049 Madrid, Spain; 5Department of Neonatology, Beth Israel Deaconess Medical Center, Harvard Medical School, 330 Brookline Avenue, Boston, MA 02215, USA

**Keywords:** long-chain polyunsaturated fatty acids, oleic acid, palmitic acid, healthy food pyramid, breastfeeding

## Abstract

In lactating women, breast milk (BM) fatty acids may come from the diet or stored adipose tissue. Our objective was to evaluate the influence of the adherence to the healthy food pyramid (HFP), the dietary pattern in the Mediterranean region, and the maternal body composition on the BM fatty acids pattern. Fifty breastfeeding women answered a socioeconomic survey and the adherence to the HFP questionnaire (AP-Q). In addition, they provided a BM sample at 7 ± 1, 14 ± 1, and 28 ± 1 days postpartum. The body’s composition was analyzed at days 7 and 28 by bioimpedance. The BM fatty acids were analyzed by gas chromatography–mass spectroscopy. We found a negative association between the consumption of olive oil and the BM palmitic acid levels (β = −3.19 ± 1.40; *p* = 0.030), and the intake of cereals and legumes was positively associated with the BM saturated fatty acids (β = 11.48 ± 3.87; *p* = 0.005). The intake of proteins and vegetables was positively associated with the omega-3 fatty acids and negatively with the omega-6:omega-3 ratio in BM. A negative association between the maternal age (β = −0.43 ± 0.11; *p* = 0.001) and the α-linolenic acid (ALA) levels was observed, being overall AP-Q positively associated with the ALA levels (β = 0.39 ± 0.15; *p* = 0.016). Physical activity reduced both the omega-3 and omega-6 fatty acids in BM. Diet had a larger influence than the maternal body’s composition on BM fatty acids during the first month of lactation, demonstrating a better adherence to the HFP and positively impacting on the omega-3 content in BM, a fact that is modulated by one’s maternal age.

## 1. Introduction

Breast milk (BM) is the best food for infant nutrition, making its macronutrients and bioactive properties indisputable for the health of the newborn. The fat content in human BM ranges from 3.5% to 4.5%, with triglycerides as the main lipid fraction, accounting for around 95% of the total lipids. Approximately half of the fatty acids in BM are saturated (SFAs) and from them, palmitic acid (PA) represents about 23–25% of the total fatty acids. Regarding monounsaturated fatty acids (MUFAs), oleic acid (OA) constitutes 36% of the total fatty acids in BM [1]. The content of long-chain polyunsaturated fatty acids (LCPUFAs) in BM is about 15% of the total lipids [2]. LCPUFAs are usually clustered into two categories: omega-3 (*n*-3) and omega-6 (*n*-6), making linoleic acid (LA) and α-linolenic acid (ALA) the precursors of *n*-3 and *n*-6 LCPUFAs, respectively. The role of breast milk LCPUFAs as bioactive molecules is implicated in the regulation of platelet aggregation, and their inflammatory and immune response has been extensively studied [3,4]. 

Generally, it is assumed that the fatty acid composition of BM reflects the maternal plasma and, hence, the maternal diet [4]. Data from women in different settings confirm the important influence of the maternal diet on fatty acids and research has been mainly focused on LCPUFAs due to their important roles in infant development. In countries with Western diets, a gradual increase in LA content and decline in ALA has been observed in BM [4]. Other studies also show evidence that women with an imbalanced *n*-6:*n*-3 ratio in their diets have a lower docosahexaenoic acid (DHA) content in their BM [5]. The DHA dietary intake has been found to be positively correlated with its concentrations in blood and BM [6] and a dietary supplementation with foods rich in DHA increase its content in BM in the first month of lactation [7]. Additionally, it has been demonstrated that the type of fish consumed by lactating women influences the LCPUFA of the BM [8]. 

The Mediterranean diet, characterized by a high consumption of cereals, nuts, vegetables, and the use of extra virgin olive oil, has been shown to raise the plasma levels of oleic acid, and *n*-3 compared to *n*-6 LCPUFAs [9]. Furthermore, the intake of seafood, common in the countries with a Mediterranean diet, also increased the *n*-3 LCPUFAs [9]. There is evidence that maternal intake of LCPUFAs and MUFAs influences the fatty acids profile of BM [10]. However, information on the fatty acids content in BM from women in the Mediterranean region in relation to their diet is not sufficiently explored. Overall, the Spanish population could be an example of a country with an adherence to the Mediterranean diet. The healthy food pyramid (HFP) is a graphic representation of the Mediterranean diet developed by the Spanish Society for Community Population [11] and has been used as a reference for different Mediterranean areas and cultures [12]. An adherence to the Mediterranean Diet has well-recognized benefits on fetal development and, therefore, is recommended for pregnant women [13]. There is less available information on how this maternal dietary pattern affects the components of BM, such as fatty acids, to establish guidelines for lactating women [14]. On the other hand, the composition of the maternal body and the fat store acquired during pregnancy could also contribute to differences in the composition of BM. A meta-regression analysis suggested an association between the maternal body mass index and lipids of BM [15]. Other studies showed that maternal adipose tissue may serve as a reservoir of LCPUFAs for BM [16]. Adipose tissue and plasma are a reflection of the maternal dietary intake, but the relative influence the maternal diet and fat deposits have on each fatty acid in BM is not clear. 

Considering the above and the relevance of fatty acids for infant development, the objective of the present study was to evaluate the influence of the adherence to the Mediterranean diet and maternal body composition on the BM fatty acids pattern in breastfeeding Spanish women during the first month of lactation.

## 2. Materials and Methods

### 2.1. Cohort and Study Design

The population was recruited at the neonatal intensive care unit and obstetrics and gynecology service of Hospital Clínico San Carlos (HCSC, Madrid, Spain). Participants were enrolled between 30 September 2019 and 13 March 2020. The inclusion criterion was women who maintained lactation during their first month postpartum. The exclusion criteria were maternal alterations which could affect the composition of the breastmilk and cause a fetal metabolic abnormality. Fifty mothers anonymously and voluntarily accepted to participate in the study and signed the informed consent form. Each participant provided a BM sample at 7 ± 1, 14 ± 1, and 28 ± 1 days postpartum. It was not possible to obtain colostrum samples for ethical reasons. The study was approved by the Ethics Committee of HCSC (19/393-E). To ensure the anonymous treatment of the data, the women were assigned a blinded ID at the beginning of the study. Their data and the transcript to the database were collected by different researchers.

### 2.2. Maternal and Neonatal Variables

Maternal data were collected from electronic medical records and questionnaires including their maternal age (years), country of origin, and for those of non-Spanish origin, their number of years living in Spain, educational level, work situation, family type and persons that integrate within the family, parity, gravidity, whether they had had a previous miscarriage, and the type of reproduction (spontaneous/assisted reproduction techniques). Regarding the clinical data, the type of delivery (vaginal/*C*-section) and gestational age (weeks of gestation) were also recorded.

The following neonatal data at birth were recorded: the sex, birth weight (grams), length and head circumference (centimeters) and the associated Z-scores, and the Apgar score at 1 and 5 min.

### 2.3. Maternal Body Composition

The maternal anthropometric parameters were measured at days 7 ± 1 and 28 ± 1 postpartum. The height (cm) was measured by a stadiometer (Seca 217, TAQ sistemas medicos, Madrid, Spain) and the waist and hip circumferences (cm) were measured by an anthropometric tape with millimeter precision. The waist-to-hip index (WHI) was calculated. The body weight (kg), total fat (%), muscle mass (%), basal metabolic rate (kcal/day), and body mass index (BMI; kg/m^2^) were assessed by bioimpedance (Omron Healthcare HBF-514C Full Body Sensor W Scale, Madrid, Spain), according to the manufacturer’s instructions.

### 2.4. Maternal Adherence to the Healthy Food Pyramid

The adherence to the healthy food pyramid questionnaire (AP-Q) was assayed by a self-administered instrument at day 28 postpartum. This questionnaire measured the nutritional behavior for 1 month and was validated in the Spanish adult population, including breastfeeding women [17,18]. The AP-Q estimates the degree of adherence to the HFP, a standard healthy pattern of diet for Mediterranean countries [19]. The AP-Q consists of 27 multiple-choice items. The responses are clustered in 10 categories, including (1) the degree of physical activity, (2) healthy habits and food preparation techniques, (3) hydration, (4) the consumption of grains, seeds, and legumes, (5) the consumption of fruits, (6) the consumption of vegetables, (7) the oil type consumed (considering the use of extra virgin olive oil as being the appropriate adherence to the HFP), (8) the consumption of dairy products, (9) the consumption of animal proteins, and (10) snacks. The healthy habits and food preparation techniques category includes four dimensions (lifestyle, emotional balance, sleep hygiene, and culinary techniques), and the hydration category contains dimensions regarding the intake of water, soft drinks, wine and beers, and distilled beverages. Each category is scored on a scale of 0 to 1, with the exception of the dimensions of the soft drinks, wine and beers, and distilled beverages, which are scored from −1 to 1. Finally, an overall AP-Q score can be obtained by the sum of the categories and ranges from 0 to 10. The higher the score, the greater the adherence to the HFP. In the HFP, the food categories at the bottom of the pyramid have positive scores and the categories at the top have negative scores [11].

### 2.5. Breast Milk Collection

A total of 0.5 mL of BM were collected at three time points of the study by hand self-expression with an electric breast pump (Symphony^®^ Medela, Barcelona, Spain). The BM was collected before feeding the infant and, if available, from both breasts. To collect the samples, the mothers washed their hands and cleaned their breast with a gauze with soap and water. The collection of the BM was performed, after breastfeeding their infant, between 10:00 and 11:59 AM and transferred to a glass bottle and kept in the fridge; it was transferred to the laboratory on the same day. In the laboratory, the BM samples were stored at −80 °C until processing.

### 2.6. Breast Milk Processing and Fatty Acids Determination

The lipids of the BM were extracted using the Folch method [20], followed by the methylation and quantification by gas chromatography–mass spectroscopy (GC–MS). Briefly, 100 μL of BM was added to 400 µL of phosphate-buffered saline and 30 μg of heptadecanoic acid (17:0) to serve as the internal standard. The fatty acids from BM were isolated by the addition of chloroform–methanol (2:1 *v*/*v*). The lipid samples were centrifuged at 400× *g* for 5 min at 4 °C. The organic phase was removed, and the fatty acids were methylated and quantified with a Hewlett–Packard Series II 5890 chromatograph (GMI, Ramsey, MN, USA) coupled to a HP-5971 mass spectrometer (LabX, Midland, ON, Canada), and equipped with a Super-Cowax SP–10 capillary column (fused silica, inner = 0.10 μm, 15 m × 0.10 mm). The carrier gas flows through the central aperture and is unrestricted throughout the length of the column. This column is suitable for the analyses of fatty acid methyl esters (FAMEs). The column temperature was held at 100 °C during the injection. Then, it was raised at the rate of 30 °C per minute to 190 °C, held at 190 °C for 8 min, and then raised to 230 °C at the rate of 30 °C per minute. The column temperature was maintained at 230 °C for an additional 3 min, and it was cooled to 100 °C for the next analysis.

For the fatty acid’s quantification, the peak identification was based upon a comparison of both the retention time and mass spectra of the unknown peak to that of the known standards within the GC–MS database library. The mass of the FAMEs was determined by comparing the areas of unknown FAMEs to that of a fixed concentration of a 17:0 internal standard. The levels of fatty acids were reported in nmol%. The summatory of the SFAs, MUFAs, *n*-6, and *n*-3 LCPUFAs were reported. In addition, palmitic acid (PA), oleic acid (OA), LA, dihomo-γ-linolenic acid (DGLA), arachidonic acid (ARA), ALA, eicosapentaenoic acid (EPA), and DHA were extracted for its relevance to maternal and neonatal health. Furthermore, the *n*-6:*n*-3, LA:ALA, LA:DHA, and ARA:DHA ratios were calculated.

### 2.7. Statistical Analysis

The data analyses were performed using R software (version 4.1.1, R Core Team 2021) within RStudio (Version 1.4.1717, RStudio, Inc., Vienna, Austria) using the *rio, tidyverse, corrplot, dplyr, devtools, compareGroups, stats, ggpubr,* and *ggplot2* packages.

The data were expressed as the median and interquartile range [Q1; Q3] for the quantitative variables and the relative frequency (%) for the qualitative variables. A chi-squared test was used to detect the association between the proportions. Furthermore, the Kruskal–Wallis test for the pairwise comparison test was used to assess the differences in BMs fatty acids along the days of lactation, when the error probability (P) of the Kruskal–Wallis test was lower than 0.10; the day of lactation was considered to adjust the models. In addition, if the *p*-value was <0.05, the difference between the day of lactation and fatty acids was analyzed by the HSD-adjusted Tukey post hoc test.

The Rho–Spearman correlation was used to test the association between the BMs fatty acids, maternal anthropometry, and the adherence to the HFP as a nutritional pattern. When the *p*-value of the correlation was <0.05, linear regression models were built to study the influence of the maternal body’s composition and nutritional pattern on the BMs fatty acids, adjusted by the days of lactation, neonatal sex, neonatal Z-scores, Apgar at 5 min, and the gestational age. The coefficient (β), standard error, and *p*-value of each factor were extracted from the models. In addition, the adjusted R^2^ and generalized Akaike information criterion (AIC) for the fit models were calculated considering the estimation of the error variance by a maximum likelihood. A significance probability was established at *p* < 0.05.

## 3. Results

### 3.1. Maternal Sociodemographic, Obstetrical Characteristics and Body Composition

In our cohort, 66.0% were Spanish women with a median age of 34.0 [31.5; 37.0] years. The non-Spanish women had a median of 14.1 [9.0; 15.0] years of residence in Spain. A total of 55.1% had finished high school, 36.7% had bachelors, and 8.2% had postgraduate degrees. In our cohort, 71.4% were employed, and 83.3% had a biparental family, with 4.0 [3.7; 4.0] family members.

Regarding the obstetrical parameters, the women had 2.4 [1.0; 3.0] pregnancies, 0.0 [0.0; 1.0] miscarriages, 2.0 [1.0; 2.0] live births, and 2.0 [1.0; 2.0] deliveries. Overall, 94.0% had a spontaneous pregnancy and 6.0% conceived through assisted reproduction techniques. In the sample, 87.3% had singleton pregnancies and 12.7% had twin pregnancies. The type of delivery was vaginal in 70.9% and 29.1% had delivered via a *C*-section. There was no association between the route of delivery and single/twin pregnancies (χ^2^ = 0.228; *p* = 0.633). The median gestational age was 37.3 [33.8; 39.0] weeks gestation. We did not detect a statistical difference in the gestational age and type of reproduction (*p* = 0.683) nor single/twin pregnancy (*p* = 0.056).

We did not detect statistical differences between day 7 and 28 of lactation in any of the anthropometric parameters evaluated, neither for the BMI (day 7 = 25.8 [23.4; 29.6] kg/m^2^, day 28 = 25.7 [23.8; 29.0] kg/m^2^; *p* = 0.999), WHI (day 7 = 0.89 [0.85; 0.93], day 28 = 0.87 [0.81; 0.91]; *p* = 0.209), or body fat (day 7 = 37.9 [33.9; 42.5] %, day 28 = 39.4 [36.2; 42.7] %; *p* = 0.704).

The maternal AP-Q categories scored as follows: physical activity = 0.16 [0.0; 0.42], healthy habits = 0.6 [0.48; 0.69], hydration = 0.52 [0.31; 0.72], grains, seed, and legumes = 0.48 [0.35; 0.67], fruits = 1.0 [0.75; 1.0], vegetables = 0.76 [0.62; 0.87], oil type = 0.83 [0.33; 1.0], dairy products = 0.46 [0.41; 0.48], animal proteins = 0.54 [0.45; 0.63], and snacks = 0.62 [0.52; 0.76]. The overall AP-Q score was 5.57 [4.94; 6.60].

### 3.2. Neonatal Variables at Birth

A total of 43.6% of the neonates were females. Their weight was 2890 [1532; 3290] grams, producing a Z-score of 0.0 [−0.7; 0.4], the length was 47.0 [41.8; 50.0] centimeters, resulting in a Z-score of −0.3 [−0.9; 0.4], and the head circumference was 33.5 [31.0; 34.6] centimeters, producing a Z-score of −0.1 [−0.5; 0.5]. The Apgar score was 9.0 [7.0; 9.0] at 1 min and 10.0 [9.0; 10.0] at 5 min.

### 3.3. Differences in Breast Milk Fatty Acids at Days 7, 14 and 28 of Lactation

We found significant differences according to the day of lactation in the PA, LA, ARA, and LA:DHA ratio. The highest levels of LA and LA:DHA ratios were found at the end of the first month of lactation. However, the PA and ARA levels declined in the BM during the first two weeks of lactation (Table 1). Since we found differences in the fatty acids along the period of lactation, the day of lactation was used to adjust the models when the *p* < 0.10.

### 3.4. Correlations between Maternal Body Composition, Adherence to HFP and BM Fatty Acids

The maternal age was positively associated with the BMs levels of SFAs, EPA, and DHA, while it was negatively correlated with the LA levels, as well as the ratios ARA:DHA, LA:DHA, and *n*-6:*n*-3.

The maternal BMI correlated positively with the levels of PA, DGLA, LA:ALA, and the *n*-6:*n*-3 ratio, while it correlated negatively with the levels of ALA, EPA, and DHA. The WHI positively correlated with the PA levels and the body fat and basal metabolic rate correlated positively with the PA and DGLA levels in the BM (Figure 1).

Regarding the dimensions measuring the adherence to the HFP, the level of physical activity was negatively correlated with the PA, ARA, ALA, EPA, and DHA levels, whereas it was positively correlated with the LA:ALA and *n*-6:*n*-3 ratios.

The healthy habits and food preparation techniques were negatively correlated with SFAs, DGLA, LA:ALA, ARA:DHA, and *n*-6:*n*-3 and were positively correlated with the ALA, EPA, and DHA levels. Hydration was negatively correlated with DGLA, LA:ALA, LA:DHA, ARA:DHA, and *n*-6:*n*-3 and was positively correlated with the ALA, EPA, and DHA levels (Figure 1).

The consumption of the category cereals, legumes, and seeds was negatively correlated with OA, MUFA, and LA:ALA and was positively correlated with SFAs and ALA.

The intake of fruit was negatively correlated with the EPA and DHA levels and was positively correlated with the LA:DHA ratio. The intake of vegetables was negatively correlated with the ARA levels and ARA:DHA ratio and was positively correlated with the EPA levels.

The type of oil used (the use of extra virgin olive oil scoring the highest) in maternal nutrition was negatively correlated with the PA levels and ARA:DHA ratio. The consumption of dairy products was positively correlated with the OA, MUFAs, and EPA levels, while it had negative correlations with the SFAs and ARA:DHA ratio. The intake of animal protein was negatively correlated with the PA levels, LA:ALA, ARA:DHA, LA:DHA, and *n*-6:*n*-3 ratio. However, it was positively correlated with the ALA, EPA, and DHA levels. Snacks alone were negatively correlated with the DGLA levels. The mothers’ overall score of AP-Q was negatively correlated with the PA levels, LA:ALA, and ARA:DHA ratio and was positively correlated with the ALA and EPA levels (Figure 1).

The variables that showed a significant correlation were entered into the models to test their association with the levels of fatty acids in breast milk.

### 3.5. Association between BM Fatty Acids and Maternal Body Composition and Nutritional Habits

The linear regression models were adjusted by the gestational age, neonatal sex, Z-scores, Apgar at 5 min, and the day of lactation. The type of oil used in the diet was negatively associated with the PA levels in breast milk (β = −3.19 ± 1.40, *p* = 0.030). In addition, the consumption of cereals, legumes, and seeds was positively associated with SFAs (β = 11.48 ± 3.87, *p* = 0.005), but was negatively associated with the OA levels (β = −7.52 ± 2.15, P = 0.001) and MUFAs (β = −7.31 ± 2.45, *p* = 0.005). Hydration was negatively associated with the DGLA levels (β = −0.53 ± 0.25, *p* = 0.043), while physical activity and the consumption of vegetables were negatively associated with the ARA levels (β = −0.25 ± 0.08, *p* = 0.003; β = −0.34 ± 0.10, *p* = 0.002; respectively, Table 2).

The maternal age was positively associated with the EPA levels (β = 0.003 ± 0.001, *p* = 0.029), DHA (β = 0.02 ± 0.01, *p* = 0.002) and was negatively associated with the ARA:DHA (β = −0.07 ± 0.02, *p* = 0.004), LA:DHA (β = −1.84 ± 0.58, *p* = 0.003), and *n*-6:*n*-3 ratios (β = −0.43 ± 0.11, *p* = 0.001). Physical activity was negatively associated with the ALA levels (β = −0.90 ± 0.31, *p* = 0.007), and the healthy habits and food preparation techniques category was also negatively associated with the ALA (β = −0.80 ± 0.36, *p* = 0.036) and DHA levels (β = −0.31 ± 0.13, *p* = 0.023). Hydration was positively associated with the DHA levels (β = 0.42 ± 0.19, *p* = 0.037) and the intake of protein was negatively associated with the LA:ALA (β = −39.0 ± 10.0, *p* < 0.001), LA:DHA (β = −55.17 ± 19.17, *p* = 0.007), and *n*-6:*n*-3 ratios (β = −17.88 ± 3.73, *p* < 0.001). The overall score of the AP-Q questionnaire was positively associated with the ALA levels (β = 0.39 ± 0.15, *p* = 0.016; Table 3).

## 4. Discussion

BM is influenced by maternal factors, including one’s diet and body composition, which may also be interrelated. Among the components of BM, fatty acids are important due to their role as bioactive molecules and to their function of storers of energy. The main findings of our study were that fatty acids in BM are influenced by the maternal nutritional patterns, while they were not substantially modified by the maternal body’s composition.

During pregnancy, the maternal fat stores increase, and they may represent an important source of fatty acids [21,22]. Some studies have evidenced positive correlations between maternal adipose tissue and the PA and OA levels in BM [16], the main fat components of this fluid [2]. Our study also showed a positive correlation between the maternal body fat, BMI, waist-to-hip ratio, and PA in BM, which might be linked with a prior energy intake and the fat storage in adipose tissue during pregnancy. In addition, we have evidenced that physical activity and a higher adherence to the HFP, which are often associated with a reduced fat mass and adipose tissue deposits, were negatively correlated with the PA levels in BM. However, these relationships were lost in the adjusted models. The maternal diet can also impact on one’s fat storage, and it is possible that the influences of maternal body fat on the BMs fatty acids observed in previous works are related to the maternal dietary pattern, an issue which was not addressed in this study [16]. Our data could evidence that the maternal diet contributes more than the body’s composition to the different fatty acids in BM. This is in accordance with other studies, demonstrating the important impact of the maternal diet during lactation in populations of different origins [4,5,7,23]. We suggest that the maternal dietary pattern could module the availability of fatty acids in BM, directly or through a modification of the fat storages.

Approximately half of BMs fatty acids are SFAs, and from them, PA represents about 23–25%. PA is an essential fatty acid in BM, contributing to 10% of the dietary energy for the accumulations of an infant’s adipose tissue in the first months of their postnatal life. However, it has been proposed that excess PA has detrimental consequences for neonatal health [24]. It is worth noting the importance of the position of PA on the glycerol backbone for the absorption and transport, making the sn-2 position less favorable for digestion. In this sense, it has been reported that excess PA in BM has detrimental consequence for the infant; a high content of PA in BM may be due to the maternal diet or the use of vegetable oil-based infant formulas [24]. In our setting, where the used of extra virgin olive oil is considered to be the healthy type of oil, we found that the higher consumption of this type of oil, the lower the PA level in BM. It is possible that women with a better adherence to the HFP consume lower levels of SFAs. In a study from Canada, it was reported that the consumption of SFA-rich diets during pregnancy were positively associated with PA in BM, and a diet of a higher quality was associated with lower SFAs in BM [25]. Additionally, higher proportions of SFAs were found in BM from Swedish women consuming a higher proportion of full-fat products [26]. Therefore, our data suggest that it would be important to promote the use of healthier oils in the diets of lactating women, such as extra virgin olive oil, which is the main oil used in the Mediterranean diet.

Olive oil has been shown to have high levels of OA [27], which has essential functions, i.e., in reducing the melting of triacylglycerides and providing fluidity for the formation, transport, and metabolism of milk fat globules [28]. It has also been shown that OA can ameliorate the cytotoxic effects of high levels of PA which, in excess, can cause oxidative damage to cells [29]. Studies have reported a higher proportion of OA in BM from women with a Mediterranean diet, suggesting a higher use of olive oil in the dietary pattern [28,30]. The relationship between the type of diet and OA levels in BM were also found in Portuguese lactating women. In this study, a decrease in the MUFAs was observed in BM, mainly due to lower OA levels, in relation to recent changes in Portuguese women’s dietary patterns [31]. Based on these studies, we were surprised to find that the OA was not high in our BM samples. One of the possible explanations is that 44% of the population was of a non-Spanish origin, and even if they have been living in Spain for long periods, these women may have not acquired the Mediterranean dietary pattern.

Our data also evidenced that the higher maternal intake of grains, legumes, and seeds was associated with an increase in the SFAs in BM and decreased MUFAs. Other studies have found lower trans fatty acid levels in BM from women who frequently consumed these foods, which may be associated with better diets and a lower fast-food consumption [23]. There is evidence that a high content of carbohydrates in the diet may increase the SFAs proportion in BM [32]. This may be a possible explanation for our data regarding to the association between cereals and legumes, important sources of carbohydrates, and the SFAs content in BM. On the other hand, it is possible that culinary aspects may influence this, since legumes in Spain are usually consumed in combination with pork products (which are rich in SFAs).

During lactation, the maternal intake of *n*-3 LCPUFAs may be a problem in low socioeconomic countries, due to a limited access to protein from fish sources. However, it has been proposed that the intake of some types of vegetable oils increase ALA, as the precursor of the *n*-3 [33]. In French lactating women, with a low consumption of vegetable oils, fish, and sea foods, it has been reported that the maternal intake of *n*-3 LCPUFA does not meet the dietary recommendations for ALA, EPA, and DHA, with an impact on their BMs fatty acids [34]. Studies from Canada [35] and Chile, in women with Western dietary patterns, evidence that a low DHA diet was related to a decrease in the DHA levels in their BM [5].

Our data showed that a better fit to the Mediterranean diet, with higher AP-Q scores, was associated with higher ALA levels in BM. We also found that a good adjustment of the protein category had an impact on BM by decreasing the LA:DHA and *n*-6:*n*-3 ratios. We suggest this is related to the animal protein category of the AP-Q, which includes the intake of oily fish, which is rich in *n*-3 fatty acids [36]. It is likely that the type of animal protein consumed by the women in our cohort includes a high proportion of sea-derived products, since in our sociocultural context, oily fish are easily accessible at a low cost. The comparison of the BM from the North of Spain with different countries showed a great variability in their lipid content, especially in LA and ALA, because of divergent nutritional patterns. In contrast, the levels of SFAs, MUFAs, and *n*-6 LCPUFAs were unaffected by the intake of fish and a fish oil supplement [37].

Our data also support the importance of a balanced diet in an *n*-6:*n*-3 ratio for the LCPUFAs content in BM, not only as source of final products, but also due to the influence of competition between the precursors for the active sites of desaturase enzymes in biosynthetic pathways [38,39,40]. Thus, a high concentration of LA in the diet would produce a greater affinity for the Δ6-desaturase, the enzyme that catalyzes the first and rate-limiting step for the biosynthesis of LCPUFAs [41], limiting the access to ALA and decreasing the *n*-3 pathway synthesis. In accordance, there is also evidence that EPA can only be accumulated when the LA:ALA ratio of diet is low [4]. Therefore, it has been proposed that the LA:ALA ratio of diets should be reduced [42]. However, it must be noted that, although increased dietary ALA enhances the EPA synthesis, the synthesis of DHA from EPA is more complex and may even be inhibited if the ALA levels are drastically increased. The capability of ALA-rich diets to suppress DHA synthesis was supported by the works of Cleland [43] and has been demonstrated in human studies [44,45]. Our data indicate that an adjustment to the protein recommendations of the Mediterranean diet was associated with an increase in the *n*-3 pathway versus the *n*-6, which could be a benefit in the development of the diets of lactating women.

The HFP considers other aspects beside diet, such as physical activity and healthy lifestyle habits. In lactating women, fatty acids from the diet can be stored in the adipose tissue or transferred to the mammary gland for an incorporation into their BM. Since exercise mobilizes fatty acids from the body’s stores for energy, it is possible that the levels in BM may be modified by exercise [46]. According to our data, moderate physical activity, following the recommendations of the HFP, decreased the levels of ARA and ALA in BM. In men, exercise has been shown to decrease the ARA levels in plasma [47]. However, Bopp et al., studying lactating women, did not find an association between exercise and a decrease in the ARA levels in BM and suggests that women consuming adequate amounts of LCPUFAs and with sufficient fat stores can exercise moderately without decreasing the LCPUFAs in their BM [46]. In addition, it must be noted that there is a wide variety of training sessions in the different studies, varying from moderate to high intensities, which may account for the disparity of the results. Our questionnaire evaluated only moderate physical activity and did not include high-intensity training. The metabolic flexibility and maternal weight status may explain the inter-individual changes in the BMs lipid content in response to an acute bout of moderate intensity exercise [48].

In addition to diet and exercise, other maternal influences, such as genetic or sociodemographic factors, may influence the BMs fatty acid composition. In the study of Miliku et al., in a large cohort, the authors report modifications in a composition related to ethnic genotypic differences and maternal education, but did not find an influence of age, parity, delivery mode, or the sex of an infant [25]. However, our data indicate higher levels of *n*-3 fatty acids in older mothers. These results agree with those of Wilson et al., which observed a positive association between the maternal age and *n*-3 levels [49]. Similar trends could also be observed in pregnant women [50]. Given the associations found between the dietary pattern and fatty acids in BM, we suggest that older women may have a better adherence to HFP and a balanced diet, with this being reflected in their BM.

We were surprised to find a negative association between the category of healthy habits and the levels of *n*-3 LCPUFAs, particularly ALA and DHA. It should be considered that the healthy habits category of the AP-Q includes several dimensions, which may have contributed to this result. Among them, sleep may have contributed negatively, since the data of our group showed that breastfeeding women have the worse quality of sleep [51]. In lactating women, their sleep quality may be reduced because they must breastfeed their children several times during the night. It is possible that their quality of sleep influences the fatty acids in their BM, as previously demonstrated in plasma, where the DHA levels were associated with an earlier sleep timing [52]. Therefore, our data should be considered with caution and more research would be needed to evaluate the impact of the different aspects of this category on BM. The importance of including these dimensions in the nutrition has been established by the Iberoamerican Nutrition Foundation, which developed a new 3D pyramid of food and active healthy lifestyles in a sustainable environment, including: (1) food and nutrition, (2) physical activity and rest, and (3) education and hygiene [53]. Our data support the importance of further exploring other aspects of healthy habits, in addition to one’s diet, since they could affect the fatty acid levels, as well as other components of BM.

### Strength and Limitations

The strength of the present study was the simultaneous analysis of the body’s composition and maternal diet, which was considered in the adjusted models. Regarding the limitations, the first would be the lack of a longitudinal approach. Thus, it would be interesting to determine the dietary pattern of the mother and its association with the anthropometric and metabolic changes in different months along the period of lactation. This follow up is difficult, particularly since many women stop breastfeeding before the recommendation time, due to family–work conflicts.

On the other hand, the fat content is known to be affected by the moment of sampling being larger in hindmilk than in foremilk. The collection of our sample was performed always after the feeding of the infant and was therefore performed in the hindmilk period. Although, we did not analyze foremilk, we do not think this is relevant since it has been reported that foremilk and hindmilk have the same lipid composition, with this being mainly affected by individual characteristics [54].

Other data that could be interesting would be to study the bioactive factors present in BM, such as the inflammatory profile or appetite hormones, and their relationship with the nutritional, anthropometric, and neonatal health variables. Finally, the influences of sleep and healthy habits deserves an in-depth study, addressing the relative role of the different dimensions of this category.

## 5. Conclusions

The nutritional pattern during the first month postpartum exerts a higher influence on the composition of fatty acids in BM compared to the maternal body’s composition. Additionally, an adherence to the healthy food pyramid leads to higher levels of ALA in breast milk; particularly, an adherence to the intake of olive oil, a hallmark of the Mediterranean diet, reduces the palmitic acid content in BM. The association between the intake of protein and higher *n*-3:*n*-6 LCPUFAs is likely related to the large consumption of fish in Spain. On the other hand, the age of the women positively influences the *n*-3 content in their BM, likely due to a better adherence to the health food pyramid. The role of physical activity and sleep hygiene on the BMs fatty acids deserves a further in-depth analysis. The dietary guidelines should be adjusted for lactating women who may have specific nutritional needs.

## Figures and Tables

**Figure 1 nutrients-14-05280-f001:**
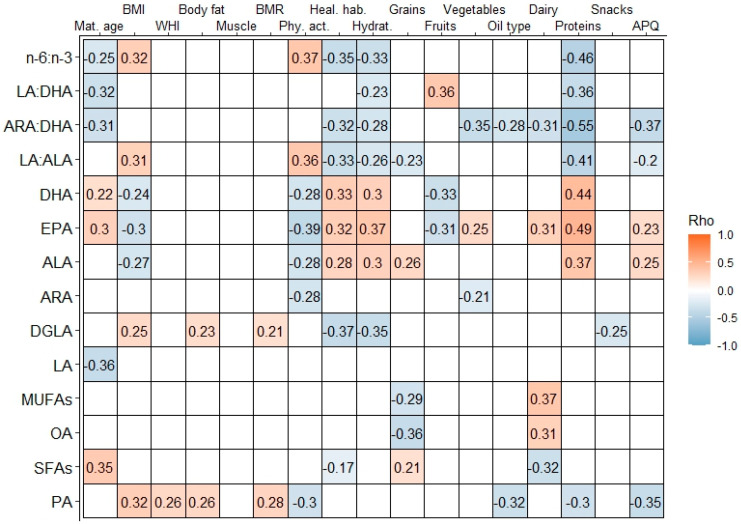
Correlogram between maternal anthropometry, healthy habits, and breast milk fatty acids. Data show Spearman (Rho) coefficient: in color when *p*-value (*p*) was lower than 0.05 (red positive correlations, blue negative correlations); otherwise, the grid is blank. PA: palmitic acid; OA: oleic acid; SFAs: saturated fatty acids; MUFAs: monosaturated fatty acids; LA: linoleic acid; DGLA: dihomo-γ-linolenic acid; ARA: arachidonic acid; ALA: α-linolenic acid; EPA: eicosapentaenoic acid; DHA: docosahexaenoic acid; *n*-6: omega-6 fatty acids: *n*-3: omega-3 fatty acids; Mat. Age: maternal age; BMI: body mass index; WHI: waist-to-hip index; BMR: Basal metabolic rate; Phy. act.: physical activity; Heal. hab.: healthy habits and food preparation techniques; Hydrat.: hydration; AP-Q: adherence to healthy pyramid questionnaire.

**Table 1 nutrients-14-05280-t001:** Breast milk fatty acids at days 7, 14, and 28 of lactation.

nmol%	Day 7 (n = 41)	Day 14 (n = 36)	Day 28 (n = 36)	*p*
Palmitic acid	19.1 [17.9; 19.9] ^a^	18.8 [17.2; 20.7] ^a^	17.1 [16.3; 19.1] ^b^	0.010
Saturated fatty acids	47.1 [43.2; 50.9]	46.8 [44.4; 50.4]	45.0 [43.0; 48.1]	0.185
Oleic acid	30.5 [27.9; 32.6]	31.7 [28.1; 33.5]	30.2 [28.6; 33.4]	0.787
Mono-saturated fatty acids	35.9 [32.3; 38.9]	35.5 [32.2; 37.7]	35.7 [33.7; 40.2]	0.890
Linoleic acid (LA)	12.3 [10.8; 14.5] ^a^	13.5 [11.4; 16.0] ^ab^	15.2 [11.7; 17.4] ^b^	0.044
Dihomo-γ-Linolenic acid	0.64 [0.59; 0.75]	0.58 [0.52; 0.74]	0.61 [0.54; 0.72]	0.129
Arachidonic acid (ARA)	0.76 [0.67; 0.85] ^a^	0.67 [0.53; 0.75] ^bc^	0.66 [0.55; 0.78] ^c^	0.007
α-Linolenic acid (ALA)	0.59 [0.47; 0.70]	0.63 [0.51; 0.78]	0.72 [0.52; 0.84]	0.096
Eicosapentaenoic acid	0.05 [0.04; 0.08]	0.06 [0.04; 0.08]	0.07 [0.04; 0.10]	0.285
Docosahexaenoic acid (DHA)	0.54 [0.42; 0.65]	0.43 [0.35; 0.55]	0.40 [0.28; 0.64]	0.071
LA:ALA	20.4 [15.3; 25.9]	21.1 [15.7; 27.2]	22.3 [14.7; 29.5]	0.914
ARA:DHA	1.43 [1.19; 1.86]	1.44 [1.17; 1.90]	1.59 [1.08; 2.35]	0.844
LA:DHA	25.0 [16.3; 29.6] ^a^	30.8 [23.6; 43.6] ^ab^	40.2 [26.2; 44.9] ^b^	0.006
*n*-6:*n*-3	11.4 [8.40; 14.3]	12.9 [9.46; 15.3]	13.3 [10.0; 17.1]	0.334

Data show median and interquartile range [Q1; Q3]. *n*-6: omega-6 fatty acids; *n*-3: omega-3 fatty acids. The *p*-value (P) extracted by Kruskal–Wallis test. Different letters indicate significant differences using the HSD-adjusted Tukey test.

**Table 2 nutrients-14-05280-t002:** Association between breast milk palmitic, oleic, SFAs, MUFAS, and omega-6 fatty acids with maternal body composition and nutritional habits.

	PA	SFAs	OA	MUFAs	LA	DGLA	ARA
Maternal age		0.06 ± 0.19 (*p* = 0.774)			−0.03 ± 0.10 (*p* = 0.749)		
BMI	−0.23 ± 0.27 (*p* = 0.399)					−0.01 ± 0.02 (*p* = 0.735)	
WHI	11.78 ± 6.49 (*p* = 0.080)						
Body fat	0.06 ± 0.17 (*p* = 0.711)					−0.01 ± 0.01 (*p* = 0.694)	
Basal metabolism	0.01 ± 0.003 (*p* = 0.175)					0.00 ± 0.00 (*p* = 0.541)	
Physical activity	−0.28 ± 1.67 (*p* = 0.866)						−0.25 ± 0.08 (*p* = 0.003)
Healthy habits		−2.30 ± 4.35 (*p* = 0.600)				−0.21 ± 0.22 (*p* = 0.350)	
Hydration						−0.53 ± 0.25 (*p* = 0.043)	
Grains		11.48 ± 3.87 (*p* = 0.005)	−7.52 ± 2.15 (*p* = 0.001)	−7.31 ± 2.45 (*p* = 0.005)			
Vegetables							−0.34 ± 0.10 (*p* = 0.002)
Oil type	−3.19 ± 1.40 (*p* = 0.030)						
Dairy products		−11.47 ± 14.71 (*p* = 0.441)	6.56 ± 8.95 (*p* = 0.469)	6.34 ± 10.19 (*p* = 0.538)			
Proteins	−5.07 ± 3.86 (*p* = 0.199)						
Snacks						−0.38 ± 0.3 (*p* = 0.136)	
AP-Q	−0.31 ± 0.70 (*p* = 0.662)						
Adjusted R^2^	0.47	0.24	0.30	0.22	0.13	0.14	0.48
AIC	188.3	272.1	236.9	248.5	333.4	−17.5	−50.3

The data show the coefficients (β) ± standard error (SE). The *p*-value (*p*) is showed in parentheses. All models were adjusted by gestational age, neonatal sex, neonatal Z-scores, and Apgar at 5 min. In addition, days of lactation was also considered when fatty acid levels showed *p* < 0.10 in the univariate analysis. The adjusted R^2^ and the Akaike information criterion (AIC) for fitted model were recorded from the models. Maternal muscle proportion and fruits category were extracted of the table due to do not shown statistical correlation with fatty acids. PA: palmitic acid; OA: oleic acid; SFAs: saturated fatty acids; MUFAs: monosaturated fatty acids; LA: linoleic acid; DGLA: dihomo-γ-linolenic acid; ARA: arachidonic acid; Mat. Age: maternal age; BMI: body mass index; WHI: waist-to-hip index; Healthy habits: healthy habits and food preparation techniques; AP-Q: adherence to healthy pyramid questionnaire.

**Table 3 nutrients-14-05280-t003:** Association between breast milk omega-3 fatty acids and ratio with maternal body composition and nutritional habits.

	ALA	EPA	DHA	LA:ALA	ARA:DHA	LA:DHA	*n*-6:*n*-3
Maternal age		0.003 ± 0.001 (*p* = 0.029)	0.02 ± 0.01 (*p* = 0.002)		−0.07 ± 0.02 (*p* = 0.004)	−1.84 ± 0.58 (*p* = 0.003)	−0.43 ± 0.11 (*p* = 0.001)
BMI	−0.002 ± 0.01 (*p* = 0.879)	−0.001 ± 0.001 (*p* = 0.423)	−0.001 ± 0.01 (*p* = 0.856)	0.06 ± 0.26 (*p* = 0.825)			0.13 ± 0.13 (*p* = 0.306)
Physical activity	−0.90 ± 0.31 (*p* = 0.007)	−0.06 ± 0.04 (*p* = 0.131)	0.09 ± 0.08 (*p* = 0.244)	7.24 ± 7.79 (*p* = 0.360)			2.30 ± 1.79 (*p* = 0.208)
Healthy habits	−0.80 ± 0.36 (*p* = 0.036)	−0.14 ± 0.08 (*p* = 0.103)	−0.31 ± 0.13 (*p* = 0.023)	−3.62 ± 9.11 (*p* = 0.694)	0.71 ± 1.68 (*p* = 0.678)		1.63 ± 2.91 (*p* = 0.580)
Hydration	0.08 ± 0.39 (*p* = 0.832)	0.02 ± 0.04 (*p* = 0.618)	0.42 ± 0.19 (*p* = 0.037)	3.83 ± 9.61 (*p* = 0.693)	−0.77 ± 0.74 (*p* = 0.306)	3.89 ± 11.00 (*p* = 0.726)	2.34 ± 3.67 (*p* = 0.528)
Grains	−0.09 ± 0.29 (*p* = 0.762)			−10.3 ± 7.04 (*p* = 0.153)			
Fruits		−0.02 ± 0.03 (*p* = 0.650)	0.09 ± 0.15 (*p* = 0.539)			17.85 ± 15.13 (*p* = 0.247)	
Vegetables		−0.03 ± 0.04 (*p* = 0.476)			0.80 ± 0.82 (*p* = 0.338)		
Oil type					−0.68 ± 0.63 (*p* = 0.293)		
Dairy products		−0.03 ± 0.16 (*p* = 0.849)			−0.11 ± 2.37 (*p* = 0.965)		
Proteins	0.12 ± 0.40 (*p* = 0.769)	−0.03 ± 0.08 (*p* = 0.683)	0.36 ± 0.18 (*p* = 0.052)	−39.0 ± 10.0 (*p* < 0.001)	−0.25 ± 1.11 (*p* = 0.824)	−55.17 ± 19.17 (*p* = 0.007)	−17.88 ± 3.73 (*p* < 0.001)
AP-Q	0.39 ± 0.15 (*p* = 0.016)	0.05 ± 0.03 (*p* = 0.109)		1.79 ± 3.81 (*p* = 0.643)	−0.58 ± 0.40 (*p* = 0.159)		
Adjusted R^2^	0.53	0.43	0.61	0.62	0.59	0.64	0.71
AIC	−3.5	−197.7	−55.2	286.0	82.0	365.5	227.0

The data show the coefficients (β) ± standard error (SE). The *p*-value (*p*) is showed in parentheses. All models were adjusted by gestational age, neonatal sex, Z-scores, and Apgar at 5 min. In addition, days of lactation was also considered when fatty acid levels showed *p* < 0.10 in the univariate analysis. The adjusted R^2^ and the Akaike information criterion (AIC) for fitted model were recorded from the models. Maternal waist-to-hip index, body fat, muscle proportion, basal metabolic rate, and snacks categories were extracted of the table due to do not shown statistical correlation with fatty acids. ALA: α-linolenic acid; EPA: eicosapentaenoic acid; DHA: docosahexaenoic acid; *n*-6: omega-6 fatty acids: *n*-3: omega-3 fatty acids; Mat. Age: maternal age; BMI: body mass index; Healthy habits: healthy habits and food preparation techniques; AP-Q: adherence to healthy pyramid questionnaire.

## Data Availability

The data presented in this study are available on request from the corresponding authors. The availability of the data is restricted to investigators based in academic institutions.

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
