# Peer review of "Association between Adherence to the Healthy Food Pyramid and Breast Milk Fatty Acids in the First Month of Lactation"

_nutrients, 2022, doi:10.3390/nu14245280_

Round 1

Reviewer 1 Report

The aim of the article is interesting and well-researched has great potential. However, there are many limitations and concerns with the manuscript and the methodology used.

Could the authors explain how it was possible to anonymously sign informed consent and then combine the data from the 3x breast milk test, the double anthropometric measurements and the questionnaire for each mother if they were anonymous?

Statistical methods are not adequate to the data collected; in the study authors obtained repeated measures for each women, for this kind of data for example linear regression model is not appropriate. Moreover, if the authors want to conclude which one - diet or maternal body fat store determined breast milk fatty acid composition it should be include in one model, not separate.

It is surprising using only newborn parameters as potential confounders in the models.

Lack of any information about models, extracting only B without any other information about the quality of model is to small amount of information.

Furthermore, if the authors wish to conclude which - maternal diet or fat stores - largely determine the fatty acid composition of maternal milk, these variables should be included in one model rather than separately.

 The results are incomprehensible described; an example: “The type of oil used in the diet was negatively associated with PA levels in breast milk” – even if the results in the table confirm this negative association, what does the type of oil used here mean?

Author Response

The aim of the article is interesting and well-researched has great potential. However, there are many limitations and concerns with the manuscript and the methodology used.

Response: Thank you for your time and comments reviewing our manuscript. We have responded your comments. In addition, the authors have discussed the suggestions and edited the text according to the recommendations.

Could the authors explain how it was possible to anonymously sign informed consent and then combine the data from the 3x breast milk test, the double anthropometric measurements, and the questionnaire for each mother if they were anonymous?

Response: During recruitment and the collection of both anthropometric and questionnaire measurements was performed by a member of the research team who knew the identity of the women. The transcription the data to the database was done by other team member with random codes blind to the rest of the research team (ID's). The databases could be linked by ID and were analyzed without knowing the original identity of the woman. This is considered anonymous from an ethical point of view. We have now included this information in the text (lines 96-98).

Statistical methods are not adequate to the data collected; in the study authors obtained repeated measures for each woman for this kind of data for example linear regression model is not appropriate. Moreover, if the authors want to conclude which one - diet or maternal body fat store determined breast milk fatty acid composition it should be include in one model, not separate.

Response: You rise two interesting points. Regarding the first, we did not analyze our data by repeated measurement because we do not have a longitudinal design for all collected variables. In our design, the AP-Q questionnaire (from which the variables of healthy and nutritional habits were extracted) and the woman's age, were only considered at one point. Secondly, the anthropometric parameters were obtained at two time points. However, possible anthropometry changes are not likely due to variations in the dietary pattern (one month is not enough for changes in nutritional habits) and are likely due to the body restructuring from pregnancy to the postnatal stage. The only variables that were collected at three time points were breast milk fatty acids, and only in those with significant changes along lactation (Table 1), the time (day) was considered as a modulating variable to adjust our models.

Regarding your second point, we have done the statistical analysis as you suggest. The models were separately constructed for each fatty acid as dependent variable. However, as you indicate, the maternal anthropometry and dietary pattern were included together in the model as independent variables.

It is surprising using only newborn parameters as potential confounders in the models.

Response: In this work, we wanted to focus on women during lactation. The variables of body composition, dietary and healthy habits were included as predictors in the models. The confounder variables were chosen based on their potential relationship with LCPUFAs, considering not only the neonatal but also the days of lactation and gestational age as obstetric and postnatal variables.

Lack of any information about models, extracting only B without any other information about the quality of model is to small amount of information.

Response: You are right. We have extracted the adjusted R2 and generalized Akaike information criterion (AIC) for fitted model. The models can be compared. This information was included in the statistical section (lines 188-189). 

Furthermore, if the authors wish to conclude which - maternal diet or fat stores - largely determine the fatty acid composition of maternal milk, these variables should be included in one model rather than separately.

Response: As previously mentioned, the predictor variables were included in one model, and not separately. They were constructed separately to predict each fatty acid, since, according to our data (Figure 1), each fatty acid is influenced by different components.

The results are incomprehensible described; an example: “The type of oil used in the diet was negatively associated with PA levels in breast milk” – even if the results in the table confirm this negative association, what does the type of oil used here mean?

Response:  In the results we did not explain every dietary item since this information about AP-Q questions has been previously published (PMID: 32503106; PMID: 34371882). In particular, “type of oil” in the AP-Q refers to the main oil type in the diet (virgin olive oil, margarines, or other types such as corn or palm oil). The higher the extra virgin olive oil consumption, the higher adherence to the healthy food pyramid. We have clarified in the methods section regarding the meaning of the categories used in AP-Q (line 126) and in the results section.

Reviewer 2 Report

BM composition changes during breastfeeding with an increase in fat usually in the end. This topic should be discussed considering that the samples were collected before starting breastfeeding.

Author Response

BM composition changes during breastfeeding with an increase in fat usually in the end. This topic should be discussed considering that the samples were collected before starting breastfeeding.

Response: Thank you for your time reviewing our manuscript. You are right, macronutrients change during the breastfeeding, and it has been shown that fat content is larger in hindmilk compared to foremilk. The collection of our sample was performed always after infant feeding, and therefore in hindmilk. We have now included this information in the method section (line 142). However, we do not think that this affected our data, since the samples were collected always at the same breastfeeding time, and it has been reported that LCPUFAs levels are almost the same between foremilk and hindmilk (PMID: 35691839). We have included this aspect in the discussion (lines 485-489).